# Total Burden of Cerebral Small Vessel Disease on MRI May Predict Cognitive Impairment in Parkinson’s Disease

**DOI:** 10.3390/jcm11185381

**Published:** 2022-09-14

**Authors:** Ruihan Zhu, Yunjing Li, Lina Chen, Yingqing Wang, Guoen Cai, Xiaochun Chen, Qinyong Ye, Ying Chen

**Affiliations:** 1Department of Neurology, Fujian Institute of Geriatrics, Fujian Medical University Union Hospital, Fuzhou 350001, China; 2Department of Neurology, The Second Affiliated Hospital, Xiamen Medical College, Xiamen 361021, China; 3Fujian Key Laboratory of Molecular Neurology, Institute of Neuroscience, Fujian Medical University, Fuzhou 350004, China; 4Institute of Clinical Neurology, Fujian Medical University, Fuzhou 350004, China

**Keywords:** Parkinson’s disease, cerebral small vessel disease burden, cognitive function

## Abstract

(1) Objective: to investigate the association between the total burden of cerebral small vessel disease (CSVD) and cognitive function in Parkinson’s disease (PD). (2) Methods: this retrospective study compared clinical and neuroimaging characteristics of 122 PD patients to determine the association between cognitive decline and total burden of CSVD in PD. All patients underwent brain MRI examinations, and their total CSVD burden scores were evaluated by silent lacunar infarction (SLI), cerebral microbleeds (CMB), white matter hyperintensities (WMH), and enlarged perivascular spaces (EPVS). The cognitive function was assessed by administering Mini-Mental State Examination (MMSE). Receiver-operating characteristic (ROC) curve and the area under the ROC curve (AUC) were performed to quantify the accuracy of the total burden of CSVD and PVH in discriminating PD patients with or without cognitive impairment. (3) Results: the PD patients with cognitive impairment had a significantly higher SLI, CMB, periventricular hyperintensities (PVH), deep white matter hyperintensities (DWMH), enlarged perivascular spaces of basal ganglia (BG-EPVS), and the total CSVD score compared with no cognitive impairment. Total CSVD score and MMSE had a significant negative correlation (r = −0. 483). Furthermore, total burden of CSVD and PVH were the independent risk factors of cognitive impairment in PD, and their good accuracy in discriminating PD patients with cognitive impairment from those with no cognitive impairment was confirmed by the results of ROC curves. (4) Conclusions: total burden of CSVD tightly linked to cognitive impairment in PD patients. The total burden of CSVD or PVH may predict the cognitive impairment in PD.

## 1. Introduction

Parkinson’s disease (PD) is a neurodegenerative disorder commonly seen in the elderly, characterized by bradykinesia, resting tremor, rigidity, and postural dysfunction. In addition to the typical motor symptoms, PD also has prominent non-motor symptoms including mood and cognitive changes, autonomic dysfunction, sleep disorders, olfactory disorders, as well as sensory abnormalities. Clinically, compared with motor symptoms, non-motor symptoms are usually more difficult to identify, and the treatment is relatively insufficient [1].

In PD, cognitive impairment (CI) is a common non-motor symptom, manifested by memory loss and executive function decline, and it severely affects patients’ quality of life. Some studies have shown that about 30–35% of PD patients have cognitive impairment [2]. Approximately 20% of new-onset PD is combined with mild cognitive impairment (MCI) [3]. More than 40% of PD patients without cognitive impairment will progress to MCI within 6 years [4], and more than 80% of PD patients will develop dementia after 20 years [5]. Once diagnosed with MCI, individuals are at a significantly increased risk of deterioration within a few years. Previous studies have shown that multiple factors are associated with cognitive decline in PD, including lower education level, disease duration, and comorbid vascular disease (diabetes, hypertension, and hyperlipidemia) [6]. There remains some research supporting the link between increasing age and male sex and the cognitive impairment observed in PD [7]. Cognitive impairment has a profound impact on the outcome of PD treatment and mortality in PD, and PD patients may die approximately 3 to 4 years after developing dementia [8,9]. Therefore, predicting and identifying cognitive impairment in PD patients early is critical to treating the disease, which may be helpful in improving their self-care abilities and prolonging their life expectancy.

Cognitive impairment is often diagnosed on the basis of clinical examination, and no reliable biomarkers have yet been identified. Studies that combine advanced structural and functional neuroimaging techniques may help to gain a better insight into the evaluation of non-motor symptoms in PD.

A disease characterized by a pathological change in the microvasculature of the brain is known as cerebral small vessel disease (CSVD) [10]. Recent studies have continued to confirm the comorbidity of Parkinson’s disease with CSVD. Cerebral hypoperfusion and neuroinflammation may underlie comorbid CSVD in Parkinson’s patients, given that these processes are involved in the progression of both Parkinson’s disease and CSVD. Approximately 76% of Parkinson’s patients may have a comorbid cerebral small vessel disease [11], which has been found to be closely associated with the progression of motor and some non-motor symptoms in PD patients [12]. Among these, comorbidity of CSVD may aggravate gait impairment and is significantly associated with depression/anxiety in PD patients. CSVD has been demonstrated to be strongly linked to cognition decline [13]. In addition, cognitive function has been reported to deteriorate as the burden of CSVD (measured by white matter lesions) increases over time [14]. It has been found that imaging markers of CSVD-related parenchymal lesions can be assessed by magnetic resonance imaging (MRI)of the brain [13]. The characteristic manifestations of CSVD on MRI include silent lacunar infarction (SLI), white matter hyperintensities (WMH), cerebral microbleeds (CMB), and enlarged perivascular spaces (EPVS) [14,15,16]. Some studies have found that SLI, as a common CSVD manifestation, could lead to cognitive impairment associated with post-infarction damage, neurofibrillary tangles, and amyloidosis [17]. Similar to SLI, several studies have found that WMH is associated with cognitive impairment, especially with worse performance in memory, executive functioning, and information processing [18]. One study reported that CMB was significantly associated with impairment in all cognitive domains except language function [19,20]. Moreover, CMB has been shown to be a risk factor for cognitive impairment in patients with subcortical vascular dementia [21]. EPVS of the brain, also known as Virchow-Robin space, is a potential fluid-filled space surrounding small vessels, and it can separate small vessels from brain parenchyma [22]. Hansen et al. reported that enlarged perivascular spaces of basal ganglia (BG-EPVS) correlate with the development of cognitive decline or dementia [23]. Furthermore, BG-EPVS may be a useful prognostic marker for cognitive decline in PD [24]. The total burden of CSVD was calculated by a cumulative score based on partial or full of manifestations of SLI, CMB, WMH, and EPVS on MRI. Most of the current studies had explored the correlation between SLI, CMB, WMH, or EPVS and cognition function. In contrast to those studies, we can gain a more intuitive and further understanding of the impact of total burden of CSVD on cognitive impairment by evaluating the total CSVD burden score, which may help guide clinical diagnosis and treatment.

Based on the relatively little research on PD cognitive impairment and the advantages of MRI, this study adopted the technology of MRI to investigate the impact of total burden of CSVD as well as CSVD-related MRI markers on cognitive function in Parkinson’s disease, in the hope that it can provide support for the prediction of total burden of CSVD in cognitive impairment of PD patients.

## 2. Materials and Methods

### 2.1. Patient Selection

This is a retrospective monocenter study from the Department of Neurology, Fujian Medical University Union Hospital, from October 2017 to December 2019. Initially, we recruited 130 PD patients. Patients with PD were diagnosed according to The International Parkinson and Movement Disorder Society (MDS)Task Force on the Definition of PD in 2016 [25]. If one or more of the following conditions existed, patients were excluded:(a)the presence of atypical, secondary, or hereditary parkinsonian syndromes.(b)head trauma history.(c)strokes associated with large vessel diseases (atherothrombotic strokes).(d)MRI contraindications.(e)incapable of completing assessments.(f)evidence of a brain tumor or hydrocephalus on MRI imaging.

Patients who had the psychiatric illness and a history of definite encephalitis were not included. This study was approved by the Ethics Committee of Fujian Medical University Union Hospital. All patients provided informed consent to this study.

### 2.2. Clinical Assessment of PD

The clinical assessments were conducted by well-trained movement disorder neurologists blind to the MRI results. The participants’ demographic information and clinical histories were recorded. The sociodemographic information included age, sex, and level of education (defined illiterate as level 0, elementary school as level 1, junior high school as level 2, high school or secondary school as level 3, college or university and above as level 4). The clinical histories included the age of onset, the duration of PD (from the occurrence of motor symptoms until the end of imaging and cognitive testing), symptoms and progression of PD and the vascular risk factors of the history of hypertension (defined by regular use of antihypertensive agents or a systolic pressure greater than 140 mmHg or a diastolic pressure greater than 90 mmHg, as demonstrated by repeated examinations), diabetes (defined by the self-reported history of diabetes or fasting blood glucose ≥ 126 mg/dL), and smoking (currently or previously smoking). The severity of PD was assessed using the Hoehn Yahr (H-Y) staging and Unified Parkinson’s Disease Rating Scale (UPDRS III) part III. Overall cognitive performance was assessed with a Mini-Mental State Examination (MMSE). A score of 27–30 on the MMSE indicates normal, while a score less than 27 indicates cognitive impairment [26]. To understand the clinical and imaging features of cognitive impairment in detail, the patients were divided into the PD-CI group (MMSE < 27) and PD-NCI group (MMSE ≥ 27) according to MMSE score. All patients would have their cognitive assessments and MRI examination on the same day.

### 2.3. Brain MRI Acquisition and Definition of CSVDs

All Brain MRI examinations were performed using a 3.0 T GE scanner (MR Discovery 750). The imaging protocol involved routine T1 weighted image (T1WI), T2 weighted image (T2WI), fluid-attenuated inversion recovery imaging (FLAIR), diffusion-weighted imaging (DWI), and enhanced T2*-weighted angiography imaging (SWAN), The SWAN parameters settings were: 4.8 mm layer with 0 mm layer spacing; repetition time (TR) 43.7 ms; echo time (TE) 23.9 ms; field of view (FOV) 22 × 20 mm; and matrix 44 × 44; Scan area ranged from the foramen magnum to the cranial vault.

SLI was defined as a round or ovoid fluid-filled cavity with the cerebrospinal fluid (CSF)-like signal, the diameter of which was between 3 and 15 mm, with the location in the subcortex. SLI usually appears as a hypointense lesion with a hyperintense rim on FLAIR images, DWI images show a hypointense vacuolated lesion [27]. WMH, defined as hyperintense lesions on T2- and FLAIR-weighted images, would be divided into two parts including periventricular hyperintensities (PVH)and deep white matter hyperintensities (DWMH). PVH and DWMH were rated, respectively, according to the Fazekas scale ranging from 0 to 3 from T2- and FLAIR weighted images [28,29]. (DWMH: 0 = absent, 1 = punctate foci, 2 = beginning confluence of foci, and 3 = large confluent areas). The presence of CMB was indicated by hypodense lesion (diameter < 10 mm) on SWAN [30], presenting a blank signal area on MRI T1-weighted or T2-weighted and FLAIR sequence (See Figure 1). EPVS defined as CSF-isointense lesions with an ovoid, round, or linear shape, were rated on MRI in the basal ganglia and centrum semiovale [31,32]. The number of EPVS would be stratified into five groups: 0 if no EPVS, 1 if EPVS = 1–10, 2 if EPVS = 11–20, 3 if EPVS = 21–40, and 4 if EPVS > 40.

MR images were independently evaluated by two senior neuroradiologists blinded to clinical information, and the mean score was obtained when there was a divergence. Finally, the total burden of CSVD was assessed by a total CSVD score using a validation scale of 0 to 4. One point was allocated to each of the following parameters [33]: the presence of SLI; the existence of cerebral microbleeds (CMB); the DWMH Fazekas score reached 2 or PVH Fazekas score reached 3; BG-EPVS score reached 2 or enlarged perivascular spaces of centrum semiovale (CS-EPVS) reached 3 [29]. The score 0 indicated there was no significant CSVD and 4 indicated there were 4 types of CSVD on imaging. In this study, a region of interest, enlarged perivascular spaces of the midbrain (Midbrain-EPVS), was additionally selected by us to represent the enlarged perivascular spaces of midbrain substantia nigra and its surroundings, which were classified into two grades: 0 (no enlarged perivascular spaces of the substantia nigra and its surroundings) and 1 (enlarged perivascular spaces of the substantia nigra and its surroundings).

### 2.4. Statistical Analysis

We used the statistical package SPSS 24.0 (SPSS Inc., Chicago, IL, USA) and GraphPad prism 7.00 (GraphPad Software, San Diego, CA, USA) to perform statistical analyses. Continuous variables were expressed as mean and standard deviations (mean ± standard deviation), and categorical variables were presented as numbers and percentages. We compared the clinical and neuroimaging characteristics of patients using the *t*-test for normally distributed continuous variables and the chi-square test for categorical variables. Spearman correlation analysis was performed to assess the association between clinical characteristics and CSVD variables. Multivariable binary logistic regression was used to identify the independent risk factors of cognitive impairment. At the same time, we further used the false discovery rate (FDR) corrected *p* < 0.05 for multiple testing. To quantify the prediction efficacy of the total burden of CSVD and PVH in discriminating PD patients with no cognitive impairment from those with cognitive impairment, the receiver operating characteristic (ROC) curve analyses, the area under the ROC curve (AUC), the sensitivity, specificity, and cut-off values were calculated, with comparisons using the DeLong’s test. ROC curves were generated using the ggplot2 package in R. The results were considered statistically significant if *p* < 0.05.

## 3. Results

### 3.1. The Clinical and Characteristics of the Study Population

Figure 2 shows the procedure for the selection of PD patients. Among them, 8 patients were excluded for unfinished MRI scans due to involuntary movements of the head and limb or claustrophobia. Finally, full clinical, cognitive, and MRI data were collected and analyzed for a total of 122 patients according to inclusion and exclusion criteria. The demographic information, clinical characteristics, and CSVD-related characteristics of patients in the two groups were presented in Table 1. As shown in Table 1, PD patients with cognitive impairment showed significant differences in age, MMSE, education level, Hoehn-Yahr staging, and hypertension compared to PD patients without cognitive impairment.

### 3.2. CSVDs and Cognitive Impairment

The cerebral vascular imaging parameters including SLI, CMB, PVH, DWMH, CS-EPVS, BG-EPVS, Midbrain-EPVS, and total burden of CSVD were also compared between the PD-NCI group and the PD-CI group. Significant differences were detected in SLI, CMB, PVH, DWMH, BG-EPVS, and the total burden of CSVD (Figure 3). PD-CI patients had significantly greater SLI, CMB, PVH, DWMH, BG-EPVS, and the total burden of CSVD. There were no differences in CS-EPVS and Midbrain-EPVS between the two groups.

### 3.3. Cognitive Impairment and Other Influence Factors

After adjusting for covariates, multivariable binary logistic regression including all collected risk factor data showed that education level (OR 0.285, 95% CI 0.171–0.476, *p* = 0.000), total burden of CSVD (OR 2.583, 95% CI 1.342–4.969, *p* = 0.004), and PVH (OR 2.523, 95% CI 1.232–5.167, *p* = 0.011) were independently associated with cognitive impairment in PD patients. Furthermore, education level was negatively correlated with cognitive impairment. In terms of CSVD-related features, the PVH and total burden of CSVD were positively associated with cognitive impairment. (Appendix A, Figure 4).

### 3.4. Correlation Analysis of the Total Burden of CSVD and Other Factors

Spearman’s correlation analysis revealed that there was a highly significant positive correlation between the total CSVD score and age (r = 0.559), hypertension (r = 0.325) as well as H-Y stage (r = 0.192). Moreover, there is evidence of a negative correlation between the total CSVD score and both MMSE (r = −0.483) and male (r = −0.269). (Table 2). No significant difference was found between the total CSVD score and other factors including PD duration, education level, diabetes and smoking.

Spearman’s correlation analysis of PVH with gender and other clinical factors in PD subjects showed that age, hypertension, diabetes, and Hoehn-Yahr stage were positively correlated with PVH; while MMSE and male were negatively correlated with PVH; in addition, disease duration, education, and smoking were not associated with PVH (see Table 2).

However, the education level correlated to none of the clinical variables and CSVD variables (Appendix A). As shown in Table 2, Spearman’s correlation analysis revealed that there was a significant correlation between the total CSVD score and MMSE (r = −0.483, *p* = 0.000). Moreover, a significant negative correlation was found between some major CSVD markers and MMSE score, except that MMSE scores were not significantly associated with Midbrain-EPVS, CMB as well as CS-EPVS.

### 3.5. Accuracy of the Total Burden of CSVD in Detecting Cognitive Impairment

The diagnostic performance of the total burden of CSVD and PVH was evaluated using the ROC curve and AUC. Figure 5 shows the graphs of the ROC curves, the total burden of CSVD showed good accuracy in detecting cognitive impairment in PD patients (AUC = 0.747, 95%CI: 0.656–0.818), with the sensitivity of 52.2% and specificity of 92.5%. In addition, The AUC of the scores for PVH was 0. 654 (95%CI: 0.571–0.737) for discriminating between PD patients with cognitive impairment from those with no cognitive impairment, which had a sensitivity and specificity of 31.9% and 98.1%, respectively.

## 4. Discussion

This study looked into the relations between CSVD and cognitive impairment in patients with PD. The data supports a robust link between the total burden of CSVD, as reflected by the total CSVD score on MRI, and cognitive impairment in PD, measured by the MMSE scale. There have been a good number of studies of a significant correlation between CSVD and cognitive impairment, but most have focused on the influence of individually different types of cerebral small vascular disease, such as SLI, CMB, WMH or EPVS. Possible associations between the total burden of CSVD and cognitive impairment have been poorly investigated. Therefore, the total CSVD score was tried in this study. Moreover, several studies showed a significant association between total CSVD score and decreased cognitive function in vascular dementia [34] or stroke or Alzheimer’s disease [35,36]. There were relatively few studies on cognitive impairment in PD. In the present study, we collected a group of patients who were clinically diagnosed with PD. The present study found statistically significant differences in SLI, CMB, WMH, and BG-EPVS between the PD-CI and PD-NCI groups (*p* < 0.05), which was in accordance with the results of the majority of studies that have shown that BG-EPVS is associated with the development of cognitive decline or dementia [24,37], while there were no statistically significant differences in CS-EPVS between the PD-CI and PD-NCI groups (*p* > 0.05), and the specific mechanism needs to be further investigated.

A longitudinal study on factors affecting PD symptoms published by Mollenhauer et al. [38] in 2019 have shown that alcohol abuse, diabetes, hypertension, and hyperlipidemia were predictors of cognitive decline in PD by investigating various baseline predictors, suggesting that vascular risk factors are significant predictors of disease progression. Consistent with this research results, our results demonstrated that hypertension affects the cognitive function of PD patients. In contrast, our recent study did not indicate statistically significant associations between diabetes and cognitive impairment in PD, which may be attributable to the relatively small sample size of subjects with diabetes in this study. The results of the present study that age of PD patients is associated with their cognitive function are consistent with the results of the study carried out by Aarsland et al. [39]. Some studies have confirmed that the severity of PD symptoms, education, and severity of motor symptoms significantly affect cognitive function in PD [40,41]. This is consistent with the results of the statistical analysis comparing the group of PD-NCI and the PD-CI in this study.

While neurodegeneration is the leading cause of Parkinson’s-related cognitive impairment, it is important to note that other factors, such as certain comorbidities, may also affect cognitive function in Parkinson’s patients. Prevalence of Parkinson’s disease and cerebrovascular disease increases with age, and these two diseases can co-exist and interact. Several studies have shown that comorbidity of CSVD is a major concern in PD and that CSVD may exacerbate motor symptoms and non-motor symptoms, including cognitive function, in PD patients [12,42].

The mechanisms by which CSVD causes cognitive impairment in PD patients are unclear. CSVD can lead to damage to white matter fiber tracts, which can cause disruption of the complex network connecting cortical and subcortical areas of relevance [43]. Extensive damage to important white matter fiber tracts in the cranial brain and disruption of the integrity of these white matter fiber tracts may be responsible for the cognitive impairment in PD due to CSVD, and the cognitive domains impaired due to white matter injury in different brain regions are different [44]. It has been found that the structural network of brain connections changes under the influence of CSVD, and the changes are characterized by a decrease in the number of brain network connections and a decrease in the intensity of brain network connections [43,45]. Anil M. Tuladhar et al. [46] suggested that CSVD affects brain networks and leads to a decrease in their intensity and a decrease in the overall and local efficiency of brain networks. Andrew J. Lawrence et al. [47] suggested that the reduction in efficiency and strength of global and local brain networks, as well as the severity of brain network disruption, is highly consistent with the severity of CSVD, which is associated with a reduction in the capacity to integrate information of the global brain as well as the capacity for local information processing due to CSVD.

WMH is one of the typical neuroimaging manifestations of CSVD and affects the cognitive function of PD patients as a factor independent of the total burden of CSVD. Relevant studies have shown that with increased brain white matter lesions, nerve fiber destruction becomes more severe and cognitive impairment becomes more apparent. A previous study has identified that severe WMH burden increases the likelihood of cognitive impairment in the early stages of the disease and causes people suffering from Parkinson’s disease to experience accelerated cognitive decline [48]. Our current study identified that PVH and DWMH were significantly greater among PD-CI patients. DWMH may disrupt short cortical connections composed of bowed “U” shaped fibers that are highly dense in areas beneath the gray matter. PVH may affect regions containing a high density of long-association and projection fiber tracts that connect more distant cortical areas. Other studies have also shown that WMH has a heterogeneous spatial distribution in PD, with WMH occurring preferentially in periventricular regions [49]. This was consistent with the results of the present study, which concluded that the severity of PVH is a risk factor for cognitive function in PD patients.

Possible mechanisms for the association between other CSVD markers and cognitive impairment in PD patients are explained as follows. The PVS is considered to be part of the lymphatic drainage system that eliminates metabolic waste, and an enlarged PVS may lead to dysfunctional clearance of soluble proteins involved in neurodegenerative diseases, which would exacerbate the pathology of PD [50]. SLI is thought to cause nerve cell death and nerve fiber degeneration, which impairs cognition by disrupting the neural structural basis associated with cognitive function, the white matter pathway [51]. CMB, on the other hand, has the potential to disrupt regional neural pathways involved in cognitive function, leading to neural network damage and cognitive impairment [52].

A prospective study regarding CSVD found that total burden of CSVD was significantly and independently associated with age, gender, hypertension, and smoking [33,53], while the present study found that total burden of CSVD in PD patients was correlated with risk factors such as age, gender, hypertension, and H-Y staging, and spearman correlation analysis of PVH as a part of CSVD with clinical factors showed the same results. However, the association between the total burden of CSVD and H-Y staging was not significant after correction of multiple comparison with FDR (*p* > 0.05). Although the association did not reach the significant threshold, it approached significance (FDR corrected *p* = 0.061). Smoking was not correlated with either PVH or total burden of CSVD. The severity of PD motor symptoms was indicated by H-Y staging, which correlated with, and may even interact with, the severity of CSVD. After that, we will expand the sample size to continue analysis and verification. In addition, further research needed to determine the exact reasons.

The AUC has values between 0.5 and 1, and the closer the AUC is to 1, the better the diagnosis. Our study concluded that the variable total burden of CSVD has moderate accuracy in predicting cognitive impairment in PD patients (AUC = 0.737, 95%CI: 0.656–0.818) and the variable PVH has a lower predictive accuracy (AUC = 0.654, 95%CI: 0.571–0.737). DeLong’s test showed that the diagnostic efficacy of CSVD was slightly better than that of PVH in predicting cognitive impairment in PD patients, and the results were not statistically significant (*p* = 0.059).

Clinically, MMSE has obvious limitations and can be easily influenced by age, education, verbal communication, and motor symptoms of PD patients, and the scale can reflect the cognitive function of patients at the time of testing, which cannot predict the long-term cognitive development of patients effectively, especially for patients without obvious cognitive impairment. Although tremor-type PD can affect the completion of MRI examination, some patients can still complete MRI examination well after pharmacological control of motor symptoms. Our results showed that CSVD or PVH was an independent risk factor for decreased MMSE scores in PD patients, which demonstrated that PVH and CSVD could predict cognitive impairment in PD. In addition, given the inextricable and stable relationship between CSVD and cognitive impairment, the assessment of CSVD and total burden of CSVD may be able to predict the long-term cognitive function of PD patients, which can provide an alternative way of thinking for clinicians to assess cognition in PD.

## 5. Conclusions

In conclusion, this study provided further evidence that there was a significant association between total burden of CSVD and cognitive impairment in patients with PD. In addition, total burden of CSVD or PVH is possible to be a reliable predictor of the long-term cognitive development in PD patients.

## Figures and Tables

**Figure 1 jcm-11-05381-f001:**
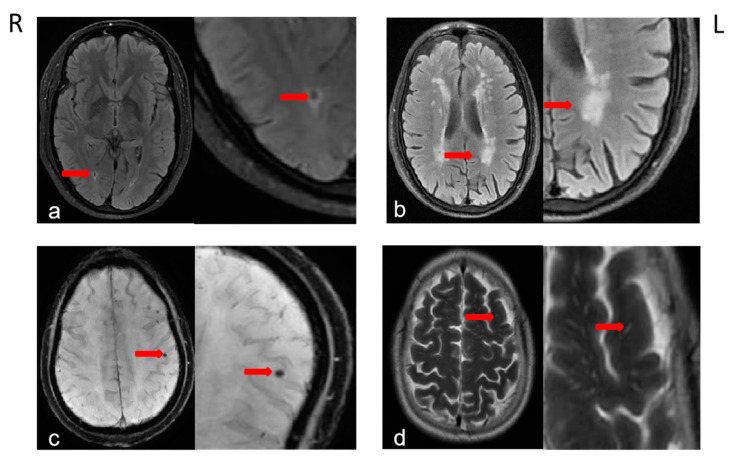
Characteristics of four CSVD-related imaging findings. The red arrow points to the location where the CSVD-related feature was examined. CSVD, cerebral small vessel disease. (**a**) SLI, silent lacunar infarction; (**b**) WMH, white matter hyperintensities; (**c**) CMB, cerebral microbleeds; (**d**) EPVS, enlarged perivascular spaces.

**Figure 2 jcm-11-05381-f002:**
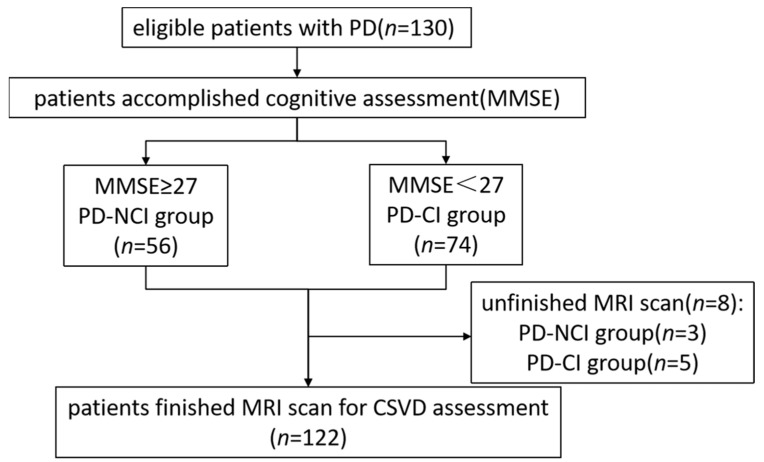
Flowchart of patient selection. PD, Parkinson's disease; MMSE, Mini-Mental State Examination; NCI, no cognitive impairment; CI, cognitive impairment; MRI, magnetic resonance imaging; CSVD, cerebral small vessel disease.

**Figure 3 jcm-11-05381-f003:**
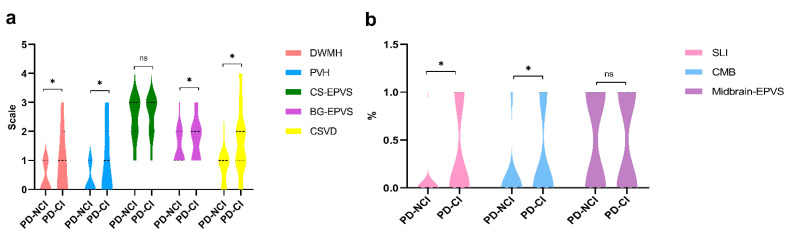
Violin plot of the comparison of the neuroimaging characteristics in two groups. (**a**) Comparison of the DWMH, PVH, CS-EPVS, BG-EPVS, CSVD in two groups; (**b**) Comparison of the SLI, CMB, Midbrain-EPVS in two groups. SLI, silent lacunar infarction; CMB, cerebral microbleeds; DWMH, deep white matter hyperintensities; PVH, periventricular hyperintensities; BG-EPVS, enlarged perivascular spaces of basal ganglia; CS-EPVS, enlarged perivascular spaces of centrum semiovale; CSVD, cerebral small vessel disease. * *p* < 0. 05; ns represents no statistical significance.

**Figure 4 jcm-11-05381-f004:**
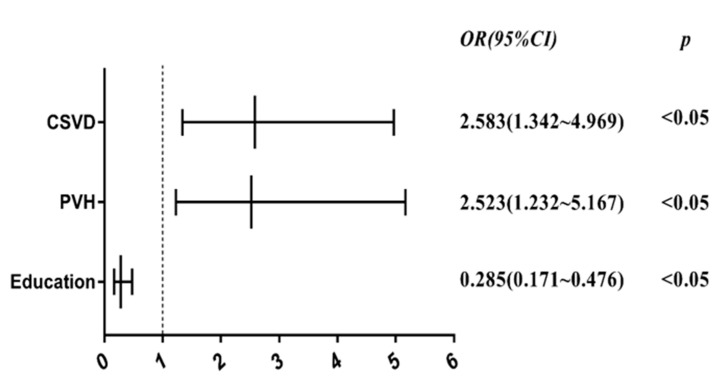
Forest plot of multivariate binary logistic regression analysis.

**Figure 5 jcm-11-05381-f005:**
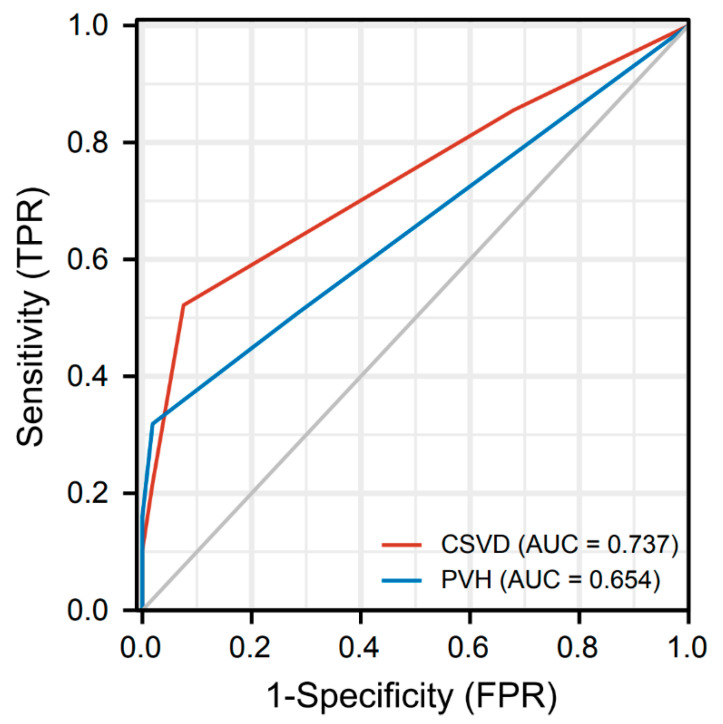
ROC Curve Analysis of the total burden of CSVD and PVH in identification of the PD patients with or without cognitive impairment. ROC: Receiver-operating characteristic; CSVD: cerebral small vessel disease; PVH: periventricular hyperintensities.

**Table 1 jcm-11-05381-t001:** Comparison of clinical and neuroimaging characteristics.

	All PD Patients (*n* = 122)	PD-NCI Group (*n* = 53)	PD-CI Group (*n* = 69)	*p*
Clinical variables				
Male ^2^	88 (72.1%)	36 (67.9%)	52 (75.4%)	0.364
Age, years ^1^	64.96 ± 8.94	61.40 ± 9.85	67.70 ± 7.10	0.000 *
MMSE score ^1^	24.45 ± 4.20	28.02 ± 0.91	21.71 ± 3.62	0.000 *
Duration, years ^1^	4.17 ± 3.64	3.53 ± 2.82	4.67 ± 4.11	0.088
Education level ^3^	1.98 ± 1.27	2.68 ± 0.976	1.45 ± 1.21	0.000 *
Hoehn-Yahr staging ^3^	2.35 ± 0.68	2.24 ± 0.66	2.45 ± 0.69	0.043 *
Hypertension ^2^	38 (31.1%)	10 (18.9%)	28 (40.6%)	0.010 *
Diabetes ^2^	13 (10.7%)	6 (11.3%)	7 (10.1%)	0.835
Smoking ^2^	22 (18.0%)	6 (11.3%)	16 (23.2%)	0.091
Imaging findings				
SLI ^1^	23 (18.9%)	1 (1.9%)	22 (31.9%)	0.000 *
CMB ^1^	24 (19.7%)	5 (9.4%)	19 (27.5%)	0.013 *
DWMH ^2^	0.69 ± 0.88	0.38 ± 0.53	0.93 ± 1.01	0.002 *
PVH ^2^	0.69 ± 0.98	0.30 ± 0.50	0.99 ± 1.14	0.001 *
CS-EPVS ^2^	2.45 ± 0.74	2.40 ± 0.77	2.49 ± 0.72	0.468
BG-EPVS ^2^	1.64 ± 0.68	1.47 ± 0.61	1.77 ± 0.71	0.019 *
Midbrain-EPVS ^1^	58 (47.5%)	26 (49.1%)	32 (46.4%)	0.769
Total CSVD score ^2^	1.30 ± 1.07	0.77 ± 0.64	1.70 ± 1.17	0.000 *

Note: Values are expressed as the mean ± SD; ^1^ Data were analyzed by two independent sample T test; ^2^ Data were analyzed by chi-square test; ^3^ Data were analyzed by rank sum test. PD, Parkinson's disease; MMSE, Mini-Mental State Examination; NCI, no cognitive impairment; CI, cognitive impairment; SLI, silent lacunar infarction; CMB, cerebral microbleeds; DWMH, deep white matter hyperintensities; PVH, periventricular hyperintensities; BG-EPVS, enlarged perivascular spaces of basal ganglia; CS-EPVS, enlarged perivascular spaces of centrum semiovale; CSVD, cerebral small vessel disease. * *p* < 0. 05.

**Table 2 jcm-11-05381-t002:** Correlation analysis of the total burden of CSVD, PVH, MMSE and clinical factors.

	The Total Burden of CSVD	PVH	MMSE
Male	−0.269 *	−0.197 *	ns
Age	0.559 *	0.546 *	−0.350 *
MMSE	−0.483 *	−0.342 *	/
Disease Duration	ns	ns	ns
Education level	ns	ns	0.538 *
Hoehn-Yahr staging	0.192 *	0.252 *	−0.277 *
Hypertension	0.325 *	0.241 *	−0.244 *
Diabetes	ns	0.274 *	ns
Smoking	ns	ns	ns
Midbrain-EPVS	/	/	ns
SLI	/	/	−0.335 *
CMB	/	/	ns
DWMH	/	/	−0.303 *
PVH	/	/	−0.324 *
CS-EPVS	/	/	ns
BG-EPVS	/	/	−0.247 *
CSVD	/	/	−0.483 *

Note: Correlation coefficients are listed. SLI, silent lacunar infarction; CMB, cerebral microbleeds; DWMH, deep white matter hyperintensities; PVH, periventricular hyperintensities; BG-EPVS, enlarged perivascular spaces of basal ganglia; CS-EPVS, enlarged perivascular spaces of centrum semiovale; CSVD, cerebral small vessel disease; MMSE, Mini-Mental State Examination. * *p* < 0. 05; ns represents no statistical significance.

## Data Availability

All data relevant to the results are included in this manuscript. Further information on the raw data can be obtained upon request from the corresponding author.

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
