# Peer review of "Total Burden of Cerebral Small Vessel Disease on MRI May Predict Cognitive Impairment in Parkinson’s Disease"

_jcm, 2022, doi:10.3390/jcm11185381_

Round 1
Reviewer 1 Report
This manuscript provides useful information on the correlation between CSVD burden and cognitive impairment in PD patients. Although the analysis is valuable and complementary to what has been published in the field, there is insufficient background information in the Introduction section. I have the following recommendations with a focus on improvements to the Introduction section.
1. Please spell out acronyms when they first show up in the manuscript. For example, PVH, DWMH, BG-EPVS were mentioned in Abstract, but what PVH, D in DWMH, and BG stand for were not explained until much later.
2. I would opt to use the term “total burden” instead of “global burden”. “Global burden” generally refers to disease burden worldwide and measures epidemiological levels.
3. In Introduction, I recommend providing more details for the following:
· What do we already know about CSVD in Parkinson patients? What percentage of Parkinson patients has CSVD? What do we know about cognitive decline in patients with CSVD and without CSVD?
· What known factors lead to or correlate with cognitive decline in Parkinson patients? The Discussion section touched on this topic, but I suggest elaborating on it in Introduction given it is very important background information.
· How is CSVD generally measured? Does that differ from how this study measures CSVD? For example, the Methods section contains the following statement: “The score 0 indicated there was no significant CSVD and 4 indicated there were 4 types of CSVD on imaging.” I suggest adding to Introduction discussing how this method used in this study compares to how CSVD is typically evaluated. When patients have low SLI, WMH scores, etc., does it mean they don’t have CSVD?
4. In Introduction, the authors discussed the associations identified by existing studies between cognitive impairment and SLI, WMH, and CMB. However, the authors described what EPVS is but not whether it has been shown to be associated with cognitive decline. I suggest commenting on this topic.
5. Why 27 was chosen as the MMSE cutoff for separating the PD patients into CI and NCI groups? Please add rationale in the manuscript.
6. The Methods section contains the following statement: “ In this study, an additional region of interest, enlarged perivascular spaces of midbrain (Midbrain-EPVS), was selected to represent the enlarged perivascular spaces of midbrain substantia nigra and its surroundings, which were classified into two grades: 0 (no enlarged perivascular spaces of the substantia nigra and its surroundings) and 1 (enlarged perivascular spaces of the substantia nigra and its surroundings). I suggest adding more clarity on how Midbrain-EPVS grade was incorporated into CSVD scoring. Does it mean the maximum score for one patient is 5 instead of 4?
7. Why were Figure 3 and 4 not combined in one plot?
8. The article requires minor English editing. For example, in line 50-51,“Once diagnosed with MCI, individuals are at a significantly increased risk of deterioration within a few years,“ the comma at the end needs to be changed to a period. In line 57-59, “Studies that combine advanced structural and functional neuroimaging techniques may help to gain a better insight into the evaluate of non-motor symptoms in PD.” “evaluate” needs to be “evaluation”.
Reviewer 2 Report
The paper was well written and easy to follow.
The result is helpful in clinic showing that global burden of CVSD correlated with cognitive impairment in PD patients and global burden of CVSD and PVH may predict cognitive impairment in PD.
I think there were too many tables in the paper. The figures were informative.
There were multiple comparisons and correlations in the paper. Did the author use multiple-comparison correction for statistical analysis?
Author Response
请参阅附件。
